# The Antioxidant Effect of Curcumin and Rutin on Oxidative Stress Biomarkers in Experimentally Induced Periodontitis in Hyperglycemic Wistar Rats

**DOI:** 10.3390/molecules26051332

**Published:** 2021-03-02

**Authors:** Gilda M. Iova, Horia Calniceanu, Adelina Popa, Camelia A. Szuhanek, Olivia Marcu, Gabriela Ciavoi, Ioana Scrobota

**Affiliations:** 1Dental Medicine Department, Faculty of Medicine and Pharmacy, University of Oradea, 410068 Oradea, Romania; gilda_iova@yahoo.ro (G.M.I.); gciavoi@uoradea.ro (G.C.); ioana_scrobota@yahoo.com (I.S.); 2Department of Periodontology Faculty of Dental Medicine, Victor Babes University of Medicine and Pharmacy, 300041 Timisoara, Romania; 3Periodontal and Periimplant Diseases Research Center “Prof. Dr. Anton Sculean”, Faculty of Dental Medicine, Victor Babes University of Medicine and Pharmacy, 300041 Timisoara, Romania; 4Department of Orthodontics, Victor Babes University of Medicine and Pharmacy, 300041 Timisoara, Romania; cameliaszuhanek@umft.ro; 5Orthodontic Research Center (ORTHO-CENTER), Faculty of Dental Medicine, Victor Babes University of Medicine and Pharmacy, 300041 Timisoara, Romania; 6Preclinics Department, Faculty of Medicine and Pharmacy, University of Oradea, 410068 Oradea, Romania; oli_baciu@yahoo.com

**Keywords:** antioxidant effect, curcumin, rutin, periodontitis, diabetes, oxidative stress

## Abstract

Background: There is a growing interest in the correlation between antioxidants and periodontal disease. In this study, we aimed to investigate the effect of oxidative stress and the impact of two antioxidants, curcumin and rutin, respectively, in the etiopathology of experimentally induced periodontitis in diabetic rats. Methods: Fifty Wistar albino rats were randomly divided into five groups and were induced with diabetes mellitus and periodontitis: (1) (CONTROL)—control group, (2) (DPP)—experimentally induced diabetes mellitus and periodontitis, (3) (DPC)—experimentally induced diabetes mellitus and periodontitis treated with curcumin (C), (4) (DPR)—experimentally induced diabetes mellitus and periodontitis treated with rutin (R) and (5) (DPCR)—experimentally induced diabetes mellitus and periodontitis treated with C and R. We evaluated malondialdehyde (MDA) as a biomarker of oxidative stress and reduced glutathione (GSH), oxidized glutathione (GSSG), GSH/GSSG and catalase (CAT) as biomarkers of the antioxidant capacity in blood harvested from the animals we tested. The MDA levels and CAT activities were also evaluated in the gingival tissue. Results: The control group effect was statistically significantly different from any other groups, regardless of whether or not the treatment was applied. There was also a significant difference between the untreated group and the three treatment groups for variables MDA, GSH, GSSG, GSH/GSSG and CAT. There was no significant difference in the mean effect for the MDA, GSH, GSSG, GSH/GSSG and CAT variables in the treated groups of rats with curcumin, rutin and the combination of curcumin and rutin. Conclusions: The oral administration of curcumin and rutin, single or combined, could reduce the oxidative stress and enhance the antioxidant status in hyperglycemic periodontitis rats.

## 1. Introduction

Diabetes is described as a “serious, chronic disease that occurs either when the pancreas does not produce enough insulin, or when the body cannot use the insulin it produces effectively” [1]. Type 1 and type 2 are the two most common forms of diabetes [2]. Type 1 diabetes is caused by the autoimmune destruction of pancreatic β cells in genetically predisposed individuals and results in severe insulin deficiency requiring insulin treatment. It is typically considered a childhood and adolescence disease, but it can occur at any age [3]. Type 2 diabetes mellitus (T2DM) is the most common endocrine disorder and has an increasing incidence worldwide. This is a serious public health concern due to the need for lifelong care, premature death and the fact that it remains incurable [4]. 

Diabetes is one of the major risk factors for periodontitis [5]. Less clear is the impact of periodontal diseases on the glycemic control of diabetes and the mechanisms by which it occurs. Inflammatory periodontal diseases may increase insulin resistance in a way similar to obesity, leading to an increase in glycemic control [6]. 

Periodontal disease is a pathological entity with a high incidence and wide range of symptoms that may develop into severe forms [7]. Periodontitis, every time, inevitably leads to significant and progressive damage to periodontal support tissue as a high-incidence disease. The conventional treatment methods for regenerating lost periodontal tissue currently consist of basic periodontal therapy, periodontal flap surgery and guided tissue regeneration [8,9,10]. The mechanism for healing is based on the formation of conjunctive epithelium, which is difficult to achieve, for the successful regeneration of functional periodontal tissue. In recent years, the technology for periodontal tissue engineering has evolved, and it is based on three aspects: seed cells (PDLSCs—human periodontal ligament stem cells), scaffolds and growth factors, which are expected to achieve functional periodontal tissue regeneration [11,12]. It has been speculated that rutin, as a typical flavonoid, might improve the characteristics of PDLSCs and enhance the regeneration ability of the periodontal tissue in an inflammatory environment [13]. Several studies have confirmed the beneficial effects of rutin in anti-inflammatory, antitumor and antioxidative activities; cell proliferation; apoptosis; differentiation and wound healing and have speculated that it may even enhance the ability of periodontal tissue to regenerate in an inflammatory environment [13,14,15,16].

Rutin is a glycoside that exhibits multiple pharmacological activities, including antidiabetic, antioxidant and anti-inflammatory activities in different models of rodents by scavenging free radicals and inhibiting lipid peroxidation, by preventing streptozotocin (STZ)-induced oxidative damage and by protecting pancreatic B cells to increase insulin secretion and lower blood glucose levels [17,18]. A chronic increase in blood glucose (associated with diabetes mellitus—DM) is a central factor in the production of reactive species that promote cellular damage and contribute to the development and progression of diabetic complications [19]. Rutin has been associated with the improvement in the glycemic control in patients with diabetes, suggesting that it might be a required nutritional supplement in these patients [20,21]. Wang et al., 2017 suggested that rutin may be a candidate agent that could be an alternative to the prevention and treatment of vascular disease in diabetes. These protective effects of rutin may have resulted from its ability to decrease the production of reactive oxygen species (ROS) and malondialdehyde (MDA) and to enhance the antioxidant enzyme activity of catalase (CAT) [22,23].

Curcumin, from the root of the turmeric plant *Curcuma longa*, is an extended pseudosymmetric polyphenol (diferuloylmethane) [24,25,26]. Zhou et al. showed that curcumin prevents bone loss in an experimental periodontitis model [27]. In association with studies that do demonstrate an effect on the prevention of bone loss, these investigations have shown that curcumin has a profound effect on inflammation by significantly reducing the development of an inflammatory infiltrates within the periodontal lesion while simultaneously stimulating an increase in the collagen content, as well as an increase in the number of fibroblastic cells within the periodontium and associated lesions when curcumin was administered daily to rats with experimentally induced periodontitis [27,28].

## 2. Results

The malondialdehyde (MDA) and oxidized glutathione (GSSG) levels were significantly decreased (*p* < 0.05) as compared to the placebo group (DPP) after administrating rutin and curcumin. The combination between curcumin and rutin (DPCR) had a more positive effect on the MDA levels compared to just rutin (DPR) and a comparable effect compared to curcumin alone (DPC). Administrating curcumin alone (DPC group) had a more positive effect reducing the GSSG levels as compared to rutin (DPR) and curcumin and rutin (DPCR) (Figure 1, Figure 2 and Figure 3).

The glutathione (GSH), GSH/GSSC (glutathione/oxidized glutathione) and catalase (CAT) activities significantly increased (*p* < 0.05) as compared to the placebo group (DPP). The administration of curcumin had a more positive effect on the GSH levels compared to the administration of curcumin and rutin (DPCR) or rutin (DPR). Rutin (DPR) had a more positive effect compared to rutin and curcumin (DPCR) on the GSH levels. 

The levels for GSH/GSSG were increased when comparing the curcumin group (DPC) with DPR and DPCR. The combination of curcumin and rutin (DPCR) had a more positive effect than rutin alone (DPR). 

The combination of curcumin and rutin (DPCR) increased the CAT levels compared to curcumin (DPC) or rutin (DPR) alone. Administrating curcumin had a more positive effect than administrating rutin.

The results of the ANOVA test (Table 1) confirmed that there were significant differences in the mean values for all five groups of rats under investigation for all variables. It could also be seen in the high values of the F test for all variables. The total sum of squares explained the variations of MDA, GSH, GSSG, GSH/GSSG and CAT due to the changes in treatment from the control group to those with diabetes mellitus and periodontitis who were not treated with curcumin, rutin or a combination of both.

The Scheffe’s test was performed to assess which antioxidant has no impact on the mean variable values presented in Table 2. 

The control group effect was statistically significantly different from any other group, regardless of whether or not the treatment was applied. There was also a significant difference between the untreated group and the three treatment groups for variables MDA, GSH, GSSG, GSH/GSSG and CAT. There was no significant difference in the mean effect between groups 3 and 4, 3 and 5 and 4 and 5. In other words, curcumin, rutin and the combination of the two had a positive effect on the treatment of the sick group of rats. 

There was no significant difference in the mean effect of the MDA, GSH, GSSG, GSH/GSSG and CAT variables in the treated groups of rats with curcumin, rutin and the combination of curcumin and rutin. These results were supported by low F values and a lower variation due to changes in the treatments (Table 3).

There was a correlation between soft tissue and blood regarding the MDA and CAT markers (Figure 4). 

The MDA levels significantly decreased as compared to the placebo group (DPP) after administrating rutin and curcumin. The combination between curcumin and rutin (DPCR) had a more positive effect on the MDA levels compared to just rutin (DPR) and a comparable effect compared to curcumin alone (DPC). The combination of curcumin and rutin (DPCR) increased the CAT levels compared to curcumin (DPC) or rutin (DPR) alone. Administrating curcumin had a more positive effect than administrating rutin.

There was a significant difference between the MDA and CAT levels for all five groups of rats when the analysis was performed at the tissue level (Table 4).

At the gingival tissue level, there were significant differences between the MDA values comparing group 1 and all the others, similar the CAT (Table 5).

## 3. Discussion

The findings of this study indicated that blood and gingival tissue MDA levels and CAT activities were corelated in all groups.

The use of animal models in periodontal research is a necessary step before clinical trials with new biomaterials and treatments are initiated. The rat is the most extensively studied rodent for periodontal diseases pathogenesis, because it is small, easy to maintain and not expensive, and the structure of the dental gingiva is quite similar to that observed in humans, with a shallow gingival sulcus and attachment of the junctional epithelium to the cementum. A surgical model for critical periodontal defects in rats has been validated, making it possible to test new biomaterials in combination with growth factors or mesenchymal stem cells [29,30]. The most widely used strains are Wistar or the Sprague-Dawley, so 50 male Wistar albino rats were used in our research.

Periodontal disease is a multifactorial disease caused by a diverse microbial flora located in the subgingival plaque. The current management of periodontal diseases focuses on reducing the microbial load attached to the teeth and restoring gingival health through surgical and nonsurgical periodontal procedures [31]. Curcumin, the major component of turmeric, has been shown to have anti-inflammatory, antimicrobial and antioxidant action [32,33,34]. In vitro studies [35,36], animal studies [27,37] and clinical studies [38,39] have demonstrated that there is a positive association between curcumin and the evolution of periodontitis.

Epidemiological studies have confirmed that diabetes is a significant risk factor for periodontitis, and the risk of periodontitis is higher if glycemic control is poor; people with poorly controlled diabetes (who are also the most at risk for other macrovascular and microvascular complications) are at an increased risk of periodontitis and alveolar bone loss [40,41].

Rutin, a citrus flavonoid glycoside found in fruits (orange, grapefruit, lemon, apples and berries), is one of the main ingredients of numerous multivitamin preparations and herbal remedies [42]. Due to its antioxidant and anti-inflammatory properties and cytoprotective actions linked to antiaging and anticancer properties, it has some documented pharmacological effects [43,44]. Rutin’s beneficial effects on the glycemic status and microvascular and macrovascular complications associated with diabetes are due to its hypolipidemic properties. The reduction in the concentration of glucose may be explained by the effect of rutin on the cell membranes and by facilitating the entry of glucose into the cells, thus decreasing the release of glucose into the blood [45].

In this study, we investigated the influence of oxidative stress and the impact of curcumin and rutin, separately and in a mixture, on the etiopathology of experimentally induced periodontitis in diabetic rats.

In order to achieve an experimental periodontitis condition faster than periodontitis normally occurs, a stainless-steel ligature with a diameter of 0.1 mm was placed around the bilateral second mandibular rat molars [46]. The placement of a ligature around the rat’s molars promotes an inflammatory challenge with the presence of neutrophils, T- and B-lymphocytes, ulceration and apical migration of the epithelial attachment [47].

There is a growing interest in the correlation between antioxidants and periodontal disease. Therefore, this research utilizes curcumin and rutin as a single antioxidant and a mixture of both of them. The use of curcumin in periodontics has been evaluated in clinical studies [38,39], animal studies [37] and in vitro studies [35].

We evaluated parameters such as MDA, GSH, GSSC, GSH/GSSC and CAT from the baseline to day 70 to determine the oxidative stress. We found that there is a significant difference between the untreated group and the three treatment groups for these variables, but there was no significant difference in the mean effect between groups DPC (treated with curcumin) and DPR (treated with rutin), DPC (treated with curcumin) and DPCR (treated with curcumin and rutin) and DPR (treated with rutin) and DPCR (treated with curcumin and rutin).

The increase in serum MDA indicated that oxidative stress incurred sufficiently could cause free radical-mediated peroxidation of the lipid component in a cell membrane; thus, MDA is a good indicator for evaluating oxidative stress in degenerative diseases such as diabetes mellitus [48]. The mean GSH, GHH/GSSC and CAT were significantly reduced in groups 2, 3, 4 and 5 compared to the control group. Tomofuji et al. [49] reported that the GSH level decreased and alveolar bone loss and polymorphic nuclear leukocyte infiltration increased in gingival tissue, and de Araújo Júnior et al. [50] reported that the MDA levels increased and GSH levels decreased in periodontitis tissue.

Animal studies and limited clinical trials have shown positive and promising results, but there are still limitations and precautions to be addressed before a wide-ranging clinical use of curcumin in periodontitis patients. Curcumin is poorly soluble in water and must be solubilized in ethanol or dimethyl sulfoxide, has low systemic bioavailability following an oral dosage and is poorly absorbed and rapidly metabolized in the gastrointestinal tract [51,52].

Further research to improve the formulations and delivery systems of curcumin and rutin is required. There are promising results for curcumin in order to increase the absorption process, to slow down the metabolism [25] and to increase its bioavailability through the use of nanoparticles [53], liposomes [54], micelles [55] and phospholipid complexes [56].

## 4. Materials and Methods

### 4.1. Chemical/Drug Used

Streptozotocin (STZ), glucose, curcumin (C), rutin (R), 2-thiobarbituric acid, o-phthalaldehyde, hydrogen peroxide and kalium phosphate buffer were purchased from Sigma Aldrich Chemicals GmbH (Munich, Germany). STZ and glucose were dissolved in distilled water. C and R were dissolved in mineral oil at 10 µmol/L.

### 4.2. Experimental Animals

Fifty male Wistar albino rats (8 weeks old and weighing a mean ± SD 220 ± 20 g) purchased from the Animal Department of the “Iuliu Hațieganu” University of Medicine and Pharmacy, Cluj-Napoca, Romania were transferred to the BIOCOM Research Center of the Department of Physiology, Cluj Napoca, Romania. The animals were housed at a temperature of 21 ± 2 °C, 70% ± 4% humidity and a less than 12-h dark/12-h light period in a controlled environment, 5 per cage. They were fed a standard pellet laboratory diet and received water ad libitum. The experiment started after a 1-week acclimatization period, and the protocol was approved by the Ethical Committee on Animal Welfare of “Iuliu Hațieganu” University of Medicine and Pharmacy nr. 172/13 June 2019, in accordance with the Guidelines on the Care and Use of Animals for Scientific Purposes, National Advisory Committee for Laboratory Animal Research, 2004. 

### 4.3. Induction of Diabetes Mellitus

On the first day after an i.m. (intramuscular) anesthesia with an injection of ketamine xylazine cocktail (100 mg/kg body weight (bw) Ketamine and 10-mg/kg-1 bw xylazine), diabetes was induced in the rats in groups 2–5. They were given by i.v. (intravenous)injection in the caudal vein STZ 30 mgkg-1 bw, followed, after 6 h, by glucose 30%, 2 mL/animal. Hyperglycemia is induced by streptozotocin (STZ) in adult rats; the mechanism of action is by selective oxidative stress destruction of pancreatic β cells. Consequently, this lowers the circulating insulin, raises the level of blood glucose and causes significant symptoms to occur [57].

The animals in group 1 only received the same volume of the control vehicle only. After 72 h, using a glucometer, glycemia was measured in the blood harvested from the caudal vein. The rats with a <300 mg/dL glycemia level were readministered STZ and glucose as stated above. 

### 4.4. Induction of Periodontitis

In order to accelerate periodontitis, plaque-accumulating devices, such as orthodontic ligatures, were placed apical to the interproximal region and pushed into the gingival sulcus around the second molars to facilitate plaque formation, as well as disrupt the gingival epithelium, enhancing osteoclastogenesis and bone loss [58,59] Placing the ligature allowed bacteria to accumulate around the ligature, leading to rapid periodontitis. The examination was conducted by moving the dental probe on all surfaces of the tooth and probing in 6 sites—3 on the buccal side and 3 on the oral side. (mesial, central and distal of the tooth). The depth of the pocket reached its peak after 15 days, and the ligature lost contact with the bottom of the periodontal pocket, which went in the apical direction, therefore reducing its effectiveness. Consequently, during the experiment, the ligature must be pushed apically. The deepest pocket had an average pocket depth of 3.1 mm. When periodontitis was confirmed, the ligatures were removed without disrupting the plaque or calculus. 

This method was considered to be a suitable model of acute disease [60]. Ligature-only placement is the most common method for periodontitis induction (72.2%) [60].

### 4.5. Group Allocation and Experimental Design

The Wistar albino rats were randomly divided into 5 groups: (1) (CONTROL)—control group, (2) (DPP)—experimentally induced diabetes mellitus and periodontitis, (3) (DPC)—experimentally induced diabetes mellitus and periodontitis treated with curcumin (C), (4) (DPR)—experimentally induced diabetes mellitus and periodontitis treated with rutin (R) and (5) (DPCR)—experimentally induced diabetes mellitus and periodontitis treated with C and R. 

After diabetes and periodontitis were installed, the rats from groups 3–5 received the treatment with C, R or combined: in group 3 was administered C, 75 mg/kg bw/rat, in group 4 was administered R 75 mg/kg bw/rat and, in group 5, in equal proportions, a combination of C and R, 75 mg/kg bw, respectively. Group 2 received the same volume of the vehicle only. The treatment was given by oral gavage every day for 10 weeks. The choice of dosage was based on data used in other similar studies in which the effects of curcumin (15–90 mg/kg) [61,62,63,64] and rutin (25–100 mg/kg) [65] on the reduction of oxidative stress parameters were investigated.

### 4.6. Blood and Tissue Sample Collection and Analyses

At the end of the experiment, after an intraperitoneal injection of a ketamine xylazine cocktail (100 mg/kg bw ketamine and 10 mg/kg bw xylazine), 5 mL of blood was drawn from the retro-orbital venous sinus of all animals, and oxidative markers were measured. Tissue samples were collected from the gingival mucosa of the ligated teeth and homologous control teeth. The samples were used to measure the biochemical oxidative markers.

The tissue fragments were homogenized with a polytron homogenizer (Brinkman Kinematica, Lucerne, Switzerland) for 3 min on ice in phosphate-buffered saline (PBS) (pH 7.4) and added at a ratio of 1:4 (*w/v*). The suspension was centrifuged at 3000× *g* and 4 °C for 5 min to prepare the cytosolic fraction. The protein content of the homogenates was measured using the Bradford method [66].

### 4.7. Determination of Oxidative Stress Biomarkers 

MDA was determined by the fluorometric method with 2-thiobarbituric acid (TBA) described by Conti et al. [67]. 

Reduced glutathione (GSH) and oxidized glutathione (GSSG) were measured fluorometrically using o-phthalaldehyde [68]. The results were expressed in nmoles/mL. 

For the MDA assessment in the gingival samples, the protein content of the homogenates was determined by the Bradford method [69] using bovine serum albumin as the standard. Homogenates of gingival tissue were heated in a boiling water bath for 1 h in 75-mM K_2_HPO_4_, pH 3, containing 10-mM TBA. After cooling, the solution was extracted with 3 mL of n-butanol in 0.6 mL of TBA. MDA was spectrofluorometrically determined in the organic phase using a synchronous technique with 534 nm of excitation and 548 nm of emission. MDA was reported as nmol/mg protein.

Catalase (CAT) activity in the homogenates and cell lysates was measured in a reaction mixture containing 10-mM hydrogen peroxide in 50-mM kalium phosphate buffer, pH 7.4. The enzyme quantity that produced an absorbance reduction of 0.43 at 25 °C per minute at 240 nm in this system was defined as one unit of catalase activity. The activity was expressed as unit/mg protein. [70]

### 4.8. Statistical Analysis

The data was statistically analyzed using ANOVA with the Scheffe’s test to compare the levels of significance between the control and experimental groups. All statistical analyses were performed using SPSS 24 software, Armonk, NY, USA. The values of *p* ≤ 0.05 were considered significant.

## 5. Conclusions

It can be concluded that the oral administration of curcumin and rutin, single or combined, could reduce oxidative stress both in gingival tissue and blood and enhance the antioxidant status in hyperglycemic periodontitis rats. Modeling oxidative stress, these two antioxidants may have an inhibitory effect on inflammation. Future research is needed to evaluate the antioxidant effects on local inflammation compared to systemic inflammation.

## Figures and Tables

**Figure 1 molecules-26-01332-f001:**
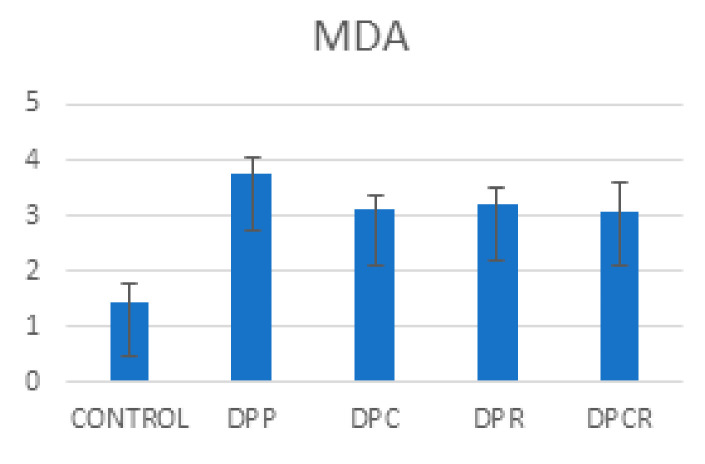
Mean variable evolution of malondialdehyde (MDA). DPP—placebo group; DPC—curcumin alone; DPR—rutin; DPCR—curcumin and rutin.

**Figure 2 molecules-26-01332-f002:**
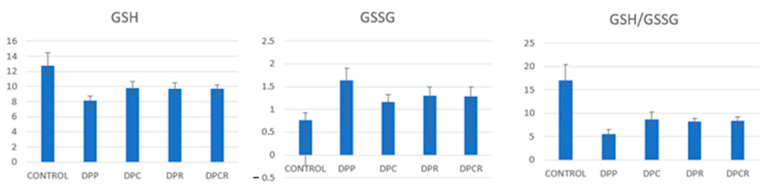
Mean variable evolution of glutathione (GSH) and oxidized glutathione (GSSG).

**Figure 3 molecules-26-01332-f003:**
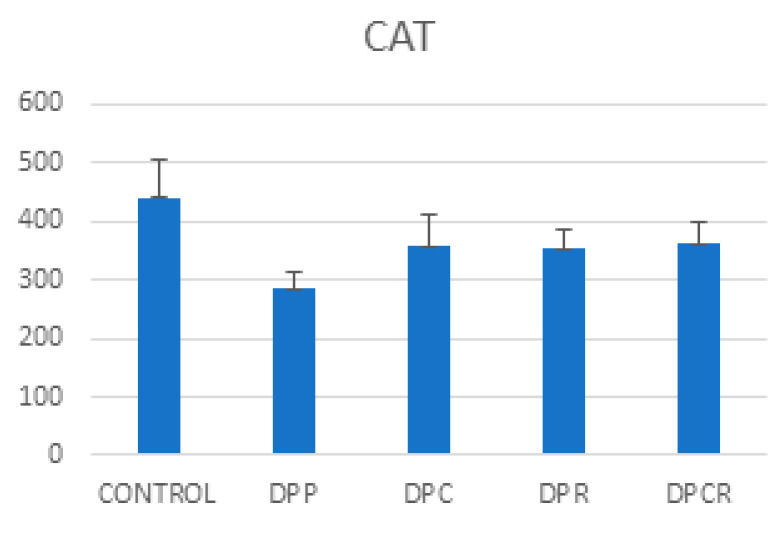
Mean variable evolution of catalase (CAT).

**Figure 4 molecules-26-01332-f004:**
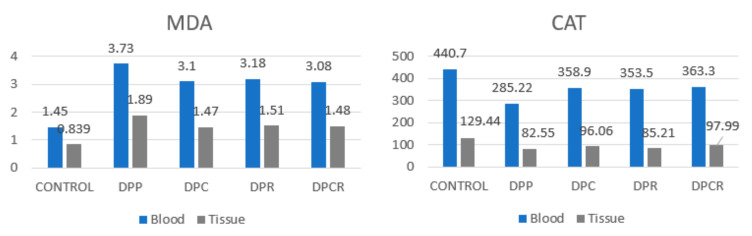
MDA and CAT markers in soft tissue and blood.

**Table 1 molecules-26-01332-t001:** Analysis of variance for all groups and all variables.

	SS	F	*p*-Value
MDA	34.97	52.366	0
GSH	150.29	26.08	0
GSSG	5.66	22.77	0
GSH/GSSG	882.56	55.26	0
CAT	206,138	14.1	0

SS—Total Sum of Squares, MDA—malondialdehyde, GSH—glutathione, GSSG—oxidized glutathione and CAT—catalase.

**Table 2 molecules-26-01332-t002:** Scheffe’s test results for multiple comparisons.

		MDA		GSH		GSSG		GSH/GSSG		CAT	
Group	Group	Mean Diff	*p*-Value	Mean Diff	*p*-Value	Mean Diff	*p*-Value	Mean Diff	*p*-Value	Mean Diff	*p*-Value
1	2	−2.28 *	0	4.53 *	0	−0.87 *	0	11.48 *	0	155.47 *	0
	3	−1.65 *	0	2.87 *	0	−0.39 *	0.004	8.35 *	0	81.8 *	0.007
	4	−1.73 *	0	3.01 *	0	−0.54 *	0	8.69 *	0	87.2 *	0.003
	5	−1.63 *	0	3.04 *	0	−0.52 *	0	8.52 *	0	77.4 *	0.012
2	1	−2.28 *	0	4.53 *	0	−0.87 *	0	11.48 *	0	155.47 *	0
	3	0.63 *	0.015	−1.66 *	0.02	0.48 *	0	−3.13 *	0.015	−74 *	0.024
	4	0.54 *	0.05	−1.51 *	0.043	0.32 *	0.026	−2.78 *	0.039	−68.27 *	0.043
	5	0.65 *	0.012	−1.49 *	0.047	0.34 *	0.016	−2.95 *	0.024	−78.07 *	0.014
3	1	−1.65 *	0	2.87 *	0	−0.39 *	0.004	8.35 *	0	81.8 *	0.007
	2	0.63 *	0.015	−1.66 *	0.02	0.48 *	0	−3.13 *	0.015	−74 *	0.024
	4	−0.08	0.991	0.14	0.999	−0.15	0.591	0.34	0.996	5.4	0.999
	5	0.013	1	0.16	0.998	−0.13	0.709	0.17	1	−4.4	1
4	1	−1.73 *	0	3.01 *	0	−0.54 *	0	8.69 *	0	87.2 *	0.003
	2	0.54 *	0.05	−1.51 *	0.043	0.32 *	0.026	−2.78 *	0.039	−68.27 *	0.043
	3	−0.08	0.991	0.14	0.999	−0.15	0.591	0.34	0.996	5.4	0.999
	5	0.1	0.984	0.02	1	0.01	1	−0.16	1	−9.8	0.993

* significant mean diff values at the significance level <0.05. Mean Diff—mean difference between the groups.

**Table 3 molecules-26-01332-t003:** Variance analysis for groups 3, 4 and 5 for all variables.

	SS	F	*p*-Value
MDA	4.28	0.19	0.825
GSH	15.64	0.14	0.862
GSSG	1.13	1.91	0.167
GSH/GSSG	32.25	0.25	0.778
CAT	45,341.37	0.14	0.866

**Table 4 molecules-26-01332-t004:** Analysis of variance of MDA and CAT for all groups.

	SS	F	*p*-Value
MDA	4.68	8.01	0
CAT	9975.88	11.6	0

**Table 5 molecules-26-01332-t005:** Scheffe’s test results for multiple comparisons.

		MDA		CAT	
Group	Group	Mean Diff	*p*-Value	Mean Diff	*p*-Value
1	2	−1.05 *	0.01	46.89 *	0.0
	3	−0.63 *	0.05	33.38 *	0.00
	4	−0.67 *	0.03	44.23 *	0.00
	5	−0.64 *	0.05	31.45 *	0.01
2	1	1.05 *	0.0	−46.89 *	0.00
	3	0.41	0.33	−13.5	0.56
	4	0.38	0.42	−2.65	0.99
	5	0.41	0.34	−15.43	0.43
3	1	0.63 *	0.05	−33.38 *	0.00
	2	−0.41	0.33	13.5	0.56
	4	−0.03	1	10.84	0.74
	5	−0.003	1	−1.93	0.99
4	1	0.67 *	0.03	−44.23 *	0.00
	2	−0.38	0.42	2.65	0.99
	3	0.03	1	−10.84	0.74
	5	0.03	1	−12.78	0.61

* significant mean diff values at the significance level <0.05. Mean Diff—mean difference between the groups.

## Data Availability

The study did not report any data.

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
