# Peer review of "The Antioxidant Effect of Curcumin and Rutin on Oxidative Stress Biomarkers in Experimentally Induced Periodontitis in Hyperglycemic Wistar Rats"

_molecules, 2021, doi:10.3390/molecules26051332_

Round 1

Reviewer 1 Report

Overall a good paper with some interesting findings with respect to the use of antioxidant supplements curcumin and rutin to treat periodontitis using a rat diabetic model. The methods are well described and results effectively presented. However, the type of assays carried out are limited to looking effects of the antioxidant supplements on plasma markers. No examination of other tissues from the animals is presented in the paper. Apart from significantly lower oxidative stress biomarkers seen in all the treatment groups compared to those in control group and no evidence is presented of any changes in the extent or severity of the peritonitis in the animals treated with the antioxidant supplements compared to controls. This means the scope of the paper is very limited. If additional evidence was presented of effects on other issues and on the disease itself this would help to strengthen the manuscript.

Author Response

We kindly appreciate your suggestion, as a result, we analyzed MDA levels and CAT activities from the gingival margin around periodontitis induced site of the Wistar rats molars. (see lines 310-320, 327-333)

When Rutin and Curcumin are administered individually and combined, there is a correlation between these two markers in the soft tissue and the blood. (see lines 151-173) We assume that both diabetes mellitus and periodontitis have beneficial results after these antioxidants are administered.

Reviewer 2 Report

Your article is focus on Periodontital diseases improvement with  Curcumin and Rutin.
Periodontal diseases are collectively the most common diseases known to mankind and include gingivitis (in which the inflammation is confined to the gingiva, and is reversible with good oral hygiene) and periodontitis (in which the inflammation extends and results in tissue destruction and alveolar bone resorption). Diabetes has been unequivocally confirmed as a major risk factor for periodontitis. 

Your  experiments were focused on selected to  Antioxidant Effect of Curcumin and Rutin - antioxidant Effect on Oxidative Stress Biomarkers in experimentally Induced Periodontitis at hyperglycaemic models.  

In ABSTRACT
please organise your text into 4  paragraphs

Background: 
Methods:
Results: 
Conclusions:

1.Introduction:

In Introduction organise your text in 4-5  paragraphs ( You have 3) 

2.Results

Fig1 should be divided in 3 Figures ( 1- MDA, 2a,b,c- GSH. GSSG, GSH/GSSG 3- CAT) and please add 
x and y axes
bar plots and error bars with *

Input comments to Table 1,3
SS, F and changes between SS and F

Input comments in Table2
in Groups 1-5 which should be write in Table ( Input here info eg from abstract ) 

4. Materials and Methods
Please you should insert  Statistical analysis from Results to Materials and Methods as 4.8. Statistical Analysis

Author Response

Thank You for your kind suggestion. Please find our point by point answer in the attached word file. 

Reviewer 3 Report

The manuscript reports antioxidant effect of two natural compounds, curcumin and rutin, in a model of type 2 diabetes - associated periodontitis in a murine model. Although there are some data showing increased levels of oxidative stress markers during the disease and a moderate or low level of their reduction in compound-treated animals, this study has several drawbacks that do not allow drawing clear conclusions and that raise concerns about its impact.

First, antioxidant properties of both compounds are very well investigated by numerous groups using a really wide array of models. In all of them the compounds could reduce levels of markers of oxidative stress, and current understanding is that such compounds act through modulation of antioxidant Nrf2 pathway that controls expression of various glutathione-metabolizing and ROS-scavenging enzymes. So, current study represent just one more example of such activity. The authors should definitely discuss how their study contributes to the field.

Second, it should also be noted that activity of both compounds was rather weak, as exemplified by the presented figures. So another question would be about choice of dosages of both compounds for the study.

Third, the manuscript lacks description of animals that should show induction of T2D and proteodontitis.

Fifth, why the authors analyzed markers of oxidative stress just in blood and not in tissues close to the site of pathology? How can they argument that the stress resulted from proteodontitis and not from T2D? Why mice with only diabetes were not used as additional control?

Minor remarks.

Subection 2.1 should be moved to Experimental section

Y-axis should be added to each panel of Figure 1. Probably, the data should better be presented as bars, as there is no connection between all compared groups. Standard deviations and symbols showing statistical significance should be added here.

Author Response

(The authors gave the same response as above.)

Round 2

Reviewer 1 Report

The authors have  presented  additional evidence of effects on other tissues as requested. This has helped to strengthen the manuscript.